# Polyamine Seed Priming: A Way to Enhance Stress Tolerance in Plants

**DOI:** 10.3390/ijms252312588

**Published:** 2024-11-23

**Authors:** Łukasz Wojtyla, Karolina Wleklik, Sławomir Borek, Małgorzata Garnczarska

**Affiliations:** Department of Plant Physiology, Institute of Experimental Biology, Faculty of Biology, Adam Mickiewicz University, Poznań, ul. Uniwersytetu Poznańskiego 6, 61-614 Poznań, Poland; karolina.wleklik@amu.edu.pl (K.W.); slawomir.borek@amu.edu.pl (S.B.); malgorzata.garnczarska@amu.edu.pl (M.G.)

**Keywords:** abiotic stress, chilling stress, priming memory, rice (*Oryza sativa* L.), salt stress, seed biology, seed germination

## Abstract

Polyamines (PAs), such as putrescine, spermine, and spermidine, are bioactive molecules that play a vital role in plant responses to stresses. Although they are frequently applied to achieve higher levels of stress tolerance in plants, their function in seed biology is still not fully understood. PAs have been described in only a limited number of studies as seed priming agents, but most of the data report only the physiological and biochemical PA effects, and only a few reports concern the molecular mechanisms. In this review, we summarized PA seed priming effects on germination, seedling establishment, and young plant response to abiotic stresses, and tried to draw a general scheme of PA action during early developmental plant stages.

## 1. Introduction

Seed biology, and in particular the process of germination, has been of interest to researchers because efficient seed germination is important for agriculture. Seed priming is understood as a pre-sowing treatment aimed at improving seed performance by regulating the germination process. The majority of seed priming treatments are based on seed imbibition to a point where germination processes begin, but radicle emergence does not occur. This controlled hydration is then followed by re-drying the seeds so that they can be handled and stored like dry seeds but with enhanced vigor and better germination performance. Seed priming is distinguished from pre-soaking due to subsequent seed drying back to the level of the initial water content, which allows storing and transportation [1,2,3,4]. However, among the different methods described as biopriming, some of them involve seed imbibition together with bacterial inoculation of seeds without subsequent drying [5,6]. Improving overall seed vigor and performance under suboptimal or stressful conditions quickly became a goal of seed priming, alongside synchronizing seed emergence and enhancing the germination of poorly performing seeds [7]. Primed seeds promote more consistent germination by activating enzymes, repairing cells, synthesizing proteins, and enhancing antioxidant defenses in comparison to seeds that have not been primed. Seed priming induces osmolyte accumulation (including proline (Pro), glycine-betaine (GB)) by modifying the metabolic pathways’ response to their biosynthesis. Various types of priming treatments, such as water-based, PGR-based (plant growth regulator), osmotic solution-based, and chemical-based methods, are commonly utilized to improve stress tolerance in crop plants. Despite sharing the common features of partial pre-hydration and early activation of germination processes in seeds, the effectiveness of these techniques varies depending on the plant species and the applied priming method [4,8]. Seed priming is now successfully used both in commercial vegetable production [9] and in basic research as a model of pre-germinative metabolism responding to seed priming and stress conditions [10] to increase the tolerance to a wide range of stress factors, including abiotic stresses like drought [8,11], salt [12,13], and heat [14], as well as biotic stresses [15].

The priming process refers to a physiological and biochemical process where plants or seeds are exposed to a stimulus or treatment that enhances their ability to respond more effectively to subsequent stress or stimuli. This process involves exposing plants or seeds to a mild or sublethal stress or treatment that activates defense mechanisms, metabolic pathways, and gene expression without causing significant damage. As a result, when a plant or seed encounters more severe stress later, it is “primed” to respond more effectively, leading to improved tolerance, resistance, or recovery [16,17,18]. The concept of priming memory as one of the mechanisms explaining the positive effect on further plant development and growth is widely accepted [10,11,19]. Chen and Arora [20] proposed a hypothesis explaining the enhanced germination and stress tolerance observed in seeds that undergo priming. According to their hypothesis, priming-induced stress tolerance is achieved through two main strategies. The first strategy involves the activation of processes associated with germination, which facilitates the transition of dormant dry seeds into an active germinating state, thereby improving germination. The second strategy, referred to as stress imprinting or cross-tolerance, involves exposing seeds to pre-germination stress through imbibition-drying cycles and the use of various priming agents. The increased stress tolerance observed during the germination of primed seeds may result from a “priming memory”. This priming-related stress response could be due to epigenetic mechanisms such as DNA methylation and histone modification. Furthermore, the primed state can be passed on to the progeny, enhancing its stress resistance. This transgenerational priming is evident in plants grown from the seeds of primed parents, which retain the priming memory and are, therefore, better equipped to respond swiftly and effectively to stress [21]. When and where during the priming treatment the epigenetic modifications that will be inherited between generations are established needs to be better understood. The inheritance of epigenetic changes induced by stress factors leads to the need for more careful observation of visible changes, especially in the context of pronounced climatic changes and the resulting stress factors’ impact on plants.

## 2. Polyamines as Regulatory Molecules That Impact Plant Stress Responses

Polyamines (PAs) are small organic polycationic molecules that are known as protective molecules. Figure 1 schematically presents the key biochemical pathways associated with PA metabolism. The PA biosynthesis pathways described in plants derive their synthesis from amino acids (mainly arginine, ornithine, and methionine). The degradation pathways are linked to the activity of diamine oxidase (DAO) and polyamine oxidase (PAO) [22,23]. PAs are hubs promoting tolerance mechanisms by a complex signaling system and have a vital role in several physiological and metabolic processes, including antioxidative mechanisms, photosynthetic pigment protection, and hormonal interplay [22,24]. The chemical properties of PAs make them likely to interact primarily with nucleic acids and phospholipids. The direct interaction of PAs with DNA plays a role in regulating gene expression through various mechanisms such as directly interacting with nucleic acids, influencing gene transcription, facilitating translation initiation, and modulating epigenetic processes [25]. The interaction with RNA, in particular, has led to the exploration of a potential link between PAs and translation, which has emerged as one of the more established mechanisms by which PAs affect cellular and developmental processes, and which involves various levels of control—from ensuring the proper assembly of ribosomes to broadly stimulating translation, enhancing translation accuracy, and promoting the translation of specific open reading frames (ORF) by increasing ribosomal frameshifting [26]. The significance of PAs in regulating the translation process has not yet been demonstrated for seed maturation and germination. However, evidence for the involvement of thermospermine (Tspm) and spermidine (Spd) in translation regulation is compelling. Tspm helps maintain optimal translation levels of ORFs through an as-yet-undefined mechanism, while Spd stimulates translation via hypusination—a posttranslational modification involving the incorporation of the aminobutyl moiety of Spd. eIF5A is the only known protein to undergo hypusination, and this modification is critical for its activity, as confirmed in the translation of the *SACL* gene [26]. The question remains open as to whether this mechanism plays a key role in developmental processes and whether it can be modulated through exogenous PA application. Despite the regulation of translation, PAs can also exert a role in gene expression at the transcription phase. PAs are discussed as repressors of gene transcription due their role in the stabilization of highly condensed states of chromatin. This repressor effect of PAs on gene transcription can be counteracted by histone acetylation. Increased levels of PAs are linked to the enhanced acetylation level of specific lysine residues on histone H3 and H4. PAs also modulate nucleosome stability, which leads to changes in the epigenome as nucleosomes are barriers to enzymes involved in DNA methylation and demethylation. Also, PA metabolism is closely associated with the availability of S-adenosyl-methionine, a major methyl group donor in transmethylation reactions, and a substrate in Spd and spermine (Spm) biosynthesis [25]. Based on these molecular mechanisms of action, it could be assumed that exogenously applied PAs elicit physiological responses that are considered favorable under stress conditions.

While PAs may not directly scavenge reactive oxygen species (ROS), they participate in the activation of the antioxidant machinery, both enzymatic and non-enzymatic, under a variety of stresses. Exogenous Spm enhanced the water deficit tolerance of orchid (*Anoectochilus roxburghii*) by increasing the activation of catalase (CAT) and decreasing the hydrogen peroxide (H_2_O_2_) and malondialdehyde (MDA) levels of accumulation [27]. CAT is an antioxidative enzyme that catalyzes the decomposition of H_2_O_2_ to water and oxygen, whereas MDA can be considered a marker of membrane damage due to lipid peroxidation caused by ROS. Exogenous putrescine (Put) also enhanced drought stress tolerance in the seedlings of grapevine (*Vitis vinifera* L., Cabernet Sauvignon) by reducing H_2_O_2_ and O_2_^•−^ levels and increasing the activity of antioxidative enzymes such as superoxide dismutase (SOD), peroxidase (POD), and CAT, as well as the contents of non-enzymatic antioxidants like ascorbic acid (AsA) and glutathione (GSH) [28]. Similarly, pretreatment of barley (*Hordeum vulgare* L.) seeds with Spm improved their drought stress tolerance, which was manifested by reduced MDA content and enhanced CAT activity [29]. Spm also reduced H_2_O_2_ and MDA levels in barley leaves under lead (Pb) stress [30]. In aluminum (Al) stress, the application of Spm lowered O_2_^•−^, H_2_O_2_, and MDA content and stimulated SOD, POD, CAT, and ascorbate peroxidase (APX) activity in chloroplasts of rice (*Oryza sativa* L.) [31]. Moreover, in a salt-sensitive wild type of tomato (*Solanum lycopersicum* L.), Spd improved salt tolerance, which was manifested by reduced H_2_O_2_ and MDA content and lower electrolyte leakage (EL), which could be discussed as a mechanism of membrane protection [32]. The tolerance to salt combined with paraquat (herbicide) was also similarly improved in tomato plants by the application of Spm through lower H_2_O_2_ and MDA levels and decreased SOD activity [33].

The improved stress tolerance is also exerted by protecting photosynthetic pigments and structures. The photosynthetic rate was significantly higher in the *Vitis vinifera* seedlings treated with Put compared with the untreated control [28]. The study on *Oryza sativa* chloroplasts showed that exogenous Spd effectively reversed Al-induced chlorophyll losses and protected photosystem II (PSII) reaction centers and the photosynthetic electron transport chain by stabilizing PSII, thus improving photosynthetic performance and preserving the integrity and function of PSII by reducing the oxidative damage caused by Al toxicity [31]. Exogenous Spd alleviated the drought stress-induced inhibition of plant growth by improving photosynthetic performance in maize (*Zea mays* L.) seedlings, particularly by enhancing photochemical efficiency and the synthesis of ATP and by maintaining the structural stability of PSII and stimulating photochemical quenching in light-harvesting complex II (LHCII) [34]. Spd and Spm also mitigate the destructive impacts of high temperature in lettuce (*Lactuca sativa* L.) seedlings [35] and salinity in rapeseed (*Brassica napus* L.) seedlings [36] by enhancing photosynthetic capacity.

Polyamines also influence the level of plant hormones, modulating their signaling pathways, which may promote stress tolerance in particular species. The crosstalk between PAs and phytohormones in plant response to abiotic stress is widely discussed and commonly accepted [37]. In maize seedlings exposed to drought, exogenous Spd increased indoleacetic acid (IAA), zeatin riboside (ZR), and gibberellin A_3_ (GA_3_) while decreased salicylic acid (SA) and jasmonate (JA) content [34]. Exogenous Spd significantly increased abscisic acid (ABA) and IAA content but reduced ethylene (ET) emission under the hypoxia stress caused by flooding in bamboo (*Phyllostachys praecox*) seedlings [38]. Although PAs affect many processes in plant responses to stresses, only some of the regulatory mechanisms were confirmed at the gene expression level. Most of the confirmed evidence on the regulation of gene expression concerns arginine decarboxylase (ADC), which catalyzes the conversion of arginine to agmatine (precursor of Put) in plants [39], archaea, and some bacteria [40]. ADC plays a crucial role in the biosynthesis of Pro and PAs [39,40]. In *Arabidopsis thaliana*, there are two genes encoding ADC: *ADC1* and *ADC2*. *ADC1* shows a constitutive expression and is also responsive to cold and bacterial pathogen infection, while *ADC2* is upregulated by abiotic stresses such as salt, drought, cold, and wounding [41,42]. The *ADC1*, which is induced primarily in response to cold, is located in the endoplasmic reticulum, where, together with N-acetyltransferase1, it provides a pathway for the synthesis of N-acetylputrescine in *Arabidopsis thaliana* [43]. The stimulation of *ADC* gene expression upon stress is widely noted in plants’ response and was proposed to serve as a hub connecting diverse functions of different plant hormones with Put [41].

## 3. Polyamine Seed Priming as a Way to Modify Plant Metabolism and Improve Stress Resistance

As the previous section refers to exogenously applied and endogenous PAs as regulatory molecules that impact plant stress responses, this section focuses on the intricate relationship between PA seed priming and the defense responses of plants under stress conditions.

One of the seed priming methods is based on seed treatment with a solution composed of water and specific bioactive components. The most frequently used molecules in seed priming belong to compounds accumulated under stress exposure and phytohormones. Here, we focus on specific responses in plants grown from PA-primed seeds, which promote stress tolerance. The most often studied PA used for seed priming is Spd, followed by Spm. Primed seeds were exposed to different stresses, however, in the study with PAs, salt stress was the most frequently investigated, followed by chilling stress. The species that is most frequently studied to observe the effect of seed priming with PAs on stress resistance is *Oryza sativa* (Table 1). In this review, we have focused only on those studies in which the methodology explicitly states that seeds, after priming, were dried back to the level of their initial water content. Most of the research focuses on seed germination behavior, seedling establishment, growth parameters, and basic biochemical features such as changes in the content of photosynthetic pigments, carotenoids, and flavonoids, osmolyte accumulation, oxidative stress markers such as antioxidant enzyme activity, the presence of ROS, the accumulation of MDA, and the level of EL under stress conditions. Only a few studies focus on molecular changes such as the accumulation of specific proteins or the activation and inhibition of particular gene expressions. Currently, there are no experimental data specifically concerning the molecular mechanisms by which the PA seed priming causes the observed effects.

The first and most significant outcome of PA seed priming is improved germination along with enhanced seedling morphology and growth parameters. All priming methods have been shown to increase both the rate and speed of germination under stress conditions. In the process of seedling establishment and stress response, one crucial factor is photosynthesis efficiency. Increased content of photosynthetic pigments, such as chlorophylls and carotenoids, was observed under salt stress in *Brassica napus* seedlings grown from Spd- and Spm-primed seeds [44] as well as in *Cucurbita pepo* seedlings grown from Put-primed seeds [45] and Spd-primed *Oryza sativa* seedlings [46]. In some studies, reduced chlorophyll degradation in Spd-primed *Oryza sativa* under chromium [47,48] and salt stress [49] was also observed. Furthermore, increased expression of the gene encoding RuBisCO small subunit (*RbcS*) was noted in Spd- and Spm-primed *Oryza sativa* seeds under salt stress [50,51].

Abiotic stresses often disrupt plant growth and productivity by disturbing cellular homeostasis through the overproduction of ROS [52,53]. Seed priming induces antioxidant mechanisms in both seeds and seedlings, particularly under stress conditions [4,7]. Reduced ROS levels were observed in cabbage (*Brassica oleracea* L.) under chilling stress due to Spd seed priming [54], and in *Oryza sativa* exposed to heat [55] or chromium stress [47,48], as well as in *Oryza sativa* under salt stress following Spd priming [49]. Most studies report decreased ROS levels; however, an increase in H_2_O_2_ content was observed in silver maple (*Acer saccharinum*) exposed to desiccation after Spd seed priming [56]. The authors suggested that the moderate increase in H_2_O_2_ may play a role in stress signaling, contributing to enhanced stress tolerance. Additionally, an increase in CAT activity was observed. Enhanced activity of antioxidant enzymes, such as CAT, SOD, glutathione peroxidase (GPX), and APX, was reported in several studies [44,46,47,48,49,54,55,56,57,58,59,60,61,62]. In *Oryza sativa*, altered isozyme profiles of CAT, SOD, GPX, and APX were noted following Spd seed priming under salt stress [49]. In other studies performed on *Oryza sativa*, increased expression of genes encoding CAT, SOD, APX, and glutathione reductase (GR) was observed under salt stress in Spd- and Spm-primed seeds [50,51].

Oxidative stress also leads to lipid peroxidation in membranes, which is typically measured by MDA levels and membrane disruption, indicated by EL. Both parameters increase under a variety of stress conditions. However, in plants grown from primed seeds, the levels of damage were lower, as evidenced by reduced MDA accumulation and weaker EL. Lower MDA levels, indicating reduced lipid peroxidation, were observed after Spd seed priming in *Acer saccharinum* under desiccation [56] and in *Cucurbita pepo* [45], tomato (*Solanum lycopersicum* L.) [58], and tobacco (*Nicotiana tabacum* L.) [57] in response to salt and chilling stresses. Similarly, Spd-primed *Oryza sativa* seeds showed reduced MDA levels under salt stress [49,63], heat stress [55], chromium toxicity [47,48], chilling stress [59], and drought stress [61].

Under stresses that cause osmotic imbalance, such as salt, water (drought or flooding), temperature (chilling or heating), or heavy metal, the production of osmotically active substances is stimulated. Higher levels of soluble sugars were noted in Spd-primed *Oryza sativa* under various stress conditions [47,48,49,55,59,61], likely due to the increased activity of α-amylase [55,60,61] and α-glucosidase [55], as well as enhanced expression of their respective genes [55]. Compounds, such as Pro and GB, are also accumulated under osmotic stress. Increased Pro levels were observed in *Brassica napus* [44], quinoa (*Chenopodium quinoa*) [64], *Oryza sativa* [59], and *Solanum lycopersicum* [58]. Also, GB was accumulated more strongly in Spd-primed *Oryza sativa* under chilling stress [60], and the upregulation of genes encoding enzymes involved in Pro and GB biosynthesis was also noted under salt stress [50,51].

Polyamine seed priming also affects endogenous PA metabolism, leading to changes in PA content and the expression of PA metabolism genes [50,51,54,60,64]. For instance, in *Brassica oleracea*, seed treatment with Spd increased endogenous Spd levels, while Put levels decreased under chilling stress [54]. In *Nicotiana tabacum* seedlings, Put, Spd, and Spm levels were higher following Put seed priming in response to chilling [57]. Increased activity of the enzymes involved in PA biosynthesis, such as ADC, ornithine decarboxylase (ODC), and S-adenosylmethionine decarboxylase (SAMDC) (Figure 1), was observed in Spd-primed *Oryza sativa* under chilling stress [60]. The Spd priming promoted the accumulation of endogenous Spm content by upregulating spermine synthase genes *SPMS1* and *SPMS2* in *Oryza sativa* under chilling stress [59]. Modulation of the gene expression involved in PA biosynthesis (*SAMDC*, *SPDS*, and *SPMS*) and catabolism (*DAO* and *PAO*) was studied in Spd- and Spm-primed *Oryza sativa* under salt stress [50,51], showing increased endogenous Put (except in Spm-primed samples), Spd, and Spm in a salt-sensitive *Oryza sativa* cultivar, and decreased levels of these PAs in a salt-tolerant cultivar [51]. *Oryza sativa* seed priming with Spm and Spd led to a decrease in Spm and Spd content in salt-stressed seedlings, which was linked to the induced expression of the *PAO* gene responsible for PA degradation even upon the induction of genes involved in Spd and Spm biosynthesis, viz., *SAMDC*, *SPDS*, and *SPMS* via PA priming [50].

The above-mentioned biochemical and molecular changes correspond with alterations in phytohormone levels, which may represent one mechanism for molecular control, triggering different types of stress responses [65]. Only a few studies have focused on the interaction between PA seed priming and plant hormones. Increased GA and ET levels, alongside decreased ABA content, were observed in *Brassica oleracea* under chilling stress after Spd seed priming [54]. In *Oryza sativa* under chromium stress, lower ABA and higher SA levels were reported following Spd seed priming [47]. However, in response to salt stress, enhanced expression of the genes involved in ABA biosynthesis and ABA-inducible transcription factors was observed in *Oryza sativa*, stimulated by Spd and Spm seed priming [50]. Although evidence of the interaction of PAs and hormones begins to emerge, there are probably many differential relationships involved in the regulation of plant hormone biosynthesis and signaling.

The modulation of the physiological and biochemical response of plants to abiotic stresses reflects alteration in gene expression. Priming with both Spm and Spd enhanced the expression of genes encoding the antioxidants and enzymes involved in osmolytes biosynthesis pathways, the ABA biosynthesis enzyme, and ABA-inducible, stress-responsive genes including transcription factors and genes of late embryogenesis abundant (LEA) proteins [51]. By accumulating higher levels of endogenous Spm and Spd, increasing the (Spm + Spd)/Put ratio, and boosting the expression of *SAMDC*, *SPDS*, *SPMS*, and *DAO*, the salt-sensitive *Oryza sativa* cultivar IR-64 demonstrated improved adaptability under salt stress after priming with both the above-mentioned PAs. However, Spm had a stronger effect [51]. Enhanced induction of transcription factors (*TRAB-1* and *WRKY-71*), *LEA* genes, and the ABA biosynthetic enzyme (*NCED3*) gene in *Oryza sativa* seedlings indicates that Spm and Spd seed priming regulated the adaptive mechanism by enhancing transcription of the regulatory genes [51]. Thus, further progress is needed not only to identify the set of genes that are regulated by priming but also the set of genes that putatively regulate the polyamine priming response and efficiency themselves. The induction of the expression of genes from PA biosynthesis pathways together with the induction of genes for enzymes of PA degradation pathways led to a decrease in PA accumulation under stress in *Oryza sativa* [50]. However, although some transcriptomic analyses have examined the impact of PA seed priming on gene expression under stress conditions, there is still a need to clarify the exact effect of PAs in the seed priming process. For this purpose, in-depth analyses should be conducted to differentiate the gene expression levels between PA priming and hydro-priming of seeds. This becomes even more significant when considering studies that demonstrate the impact of priming on PA metabolism. Basra et al. [66] observed that osmopriming of onion (*Allium cepa* L.) seeds led to increased levels of Put and Spd, while the Spm level remained unchanged compared to unprimed seeds. In *Brassica napus*, seed priming resulted in elevated Put accumulation with no significant changes in the other PA content. Additionally, seed priming alters the balance between the free, bound, and conjugated forms of PA, which can influence their metabolic functions [67]. Studies have shown that the relative proportions of various PAs are more crucial for regulating plant response to stress than the absolute concentrations of a particular PA [68]. Therefore, special emphasis should be dedicated to the experimental design. In this context, the use of PA priming in comparison to hydro-priming as a control should provide better information than approaches in which the PA priming effect is compared to the response of dry seeds.

**Table 1 ijms-25-12588-t001:** Physiological, biochemical, and molecular effects of seed priming with polyamines to enhance stress tolerance. Abbreviations are explained in the text above the Table.

Plant Species	Stress	Priming Agent	Effect	Reference
*Acer saccharinum* L.	Mild and severe desiccation	Spd	Improved seed germination Increased content of the H_2_O_2_ Improved cell membrane integrity (decreased content of MDA and EL) Increased CAT activity Protected genome integrity (decreased level of oxidized guanine) Improved mitochondria morphology and efficiency	[56]
*Brassica napus* L. cv. SY Saveo (sensitive), Edimax CI (intermediate tolerant), and Dynastie (tolerant)	Salinity	Spd Spm	Improved seed germination and seedling growth Increased content of chlorophylls and carotenoids Reduced membrane injury index Elevated antioxidant and osmotic responses (increased content of phenolic compounds and activity of phenylalanine ammonia-lyase, increased activity of SOD and content of Pro)	[44]
*Brassica oleracea* L. var. *acephala*	Chilling	Spd	Improved seed germination and seedling growth Increased content of Spd, decreased Put content Enhanced expression of *ADC*, *SAMDC, SPDS,* and *SPMS* encoding PA metabolism enzymes Increased activity of SOD and POD and scavenging of H_2_O_2_ and O_2_^•−^ Increased content of GA_3_, ET, decreased content of ABA with corresponding changes in the gene expression	[54]
*Chenopodium* *quinoa* Willd.	Salinity	Spd Spm	Improved seed germination and seedling growth Increased content of Pro Impacted expression of selected stress-related and PA metabolism genes	[64]
*Cucurbita pepo* var. *styriaka*	Salinity	Put	Improved seed germination and seedling growth Increased content of chlorophylls and carotenoids Decreased content of Pro and MDA Increased α-amylase activity	[44]
*Nicotiana tabacum* L. cv. MSk326 (sensitive) and Honghuadajinyuan (tolerant)	Chilling	Put	Improved seed germination and seedling growth Decreased content of MDA Increased content of Put, Spd, and Spm Increased activity of CAT, SOD, APX, and POD	[57]
*Solanum lycopersicum* L.cv. Principe Borghese	Salinity	Put Spm Spd	Improved seed germination Increased content of chlorophylls Increased content of Pro, phenolic compounds, flavonoids Decreased content of MDA in Put-primed seeds Increased content of Ca^2+^ and transglutaminase activity Increased antioxidant activity (except Spm-primed seeds) reflected in activity of antioxidative enzymes, such as POD, PPO, and APX (except Put-primed seeds)	[58]
*Oryza sativa* L. ssp. *Japonica* cv. Zhegeng 100	Heat	Spd	Improved seed germination and seedling growth Higher level of soluble sugar content during the early germination stage Decreased content of H_2_O_2_ and MDA Increased activity of CAT and GR, and the expression of corresponding genes Increased activity of α-amylase and α-glucosidase, and induced expression of corresponding genes	[55]
*Oryza sativa* L. cv. Chunyou 927 (sensitive), and Yliangyou 689 (tolerant)	Chromium	Spd	Improved seed germination and seedling growth Prevented cell ultrastructure damage (evidenced by conventional chloroplast morphology, developed nuclear membrane, and mitochondria swelling alleviation) Alleviated degradation of photosynthetic pigments and improved gas exchange parameters Decreased EL, and the level of MDA, H_2_O_2_, and O_2_^•−^ Increased levels of soluble sugars and soluble proteins Reduced Cr augmentation with simultaneous enhanced uptake of Mn, Zn, Cu, P, Fe, and K Decreased level of ABA, increased level of SA Increased activity of CAT, SOD, POD, and APX Enhanced expression of stress resistance gene *NPR1* in both cultivars and *PR1* and *PR2* in sensitive cultivar	[47]
*Oryza sativa* L. cv. Chunyou 927 (sensitive)	Chromium toxicity	Spd	Improved seedling growth Alleviated degradation of photosynthetic pigments and improved gas exchange parameters Decreased EL Decreased level of MDA, H_2_O_2_, and O_2_^•−^ Increased activity of CAT, SOD, and POD Increased level of soluble sugars Reduced uptake of the Cr with simultaneous enhanced uptake of Mn, Zn, and Cu	[48]
*Oryza sativa* L. cv. Khao Dawk Mali 10	Salinity	Spd	Improved seed germination and seedling growth Increased level of O_2_^•−^ in the seedling radicle Decreased EL Decreased level of MDA	[63]
*Oryza sativa* L.	Chilling	Spd	Improved seed germination and seedling growth Decreased level of MDA Increased level of Pro and soluble sugars Enhanced activity of CAT, POD, and APX, decreased activity of SOD Decreased expression of *trehalose-6-phosphate–phosphatase* (*TPP1* and *TPP2*) genes, upregulation of *SPMS1* and *SPMS2*	[59]
*Oryza sativa* L. cv. IR-64	Salinity	Spd	Improved seedling growth Reduced osmotic imbalance Alleviated degradation of photosynthetic pigments Lowered EL Decreased content of MDA, H_2_O_2_ Improved antioxidant capacity by increased activity of CAT, SOD, GPX, and APX, and content of non-enzymatic (anthocyanin and cysteine) antioxidants Altered isozyme profile of CAT, SOD, GPX, and APX Decreased protease activity	[49]
*Oryza sativa* L. cv. Zhu Liang You 06 and Qian You No.1	Chilling	Spd	Improved seed germination and seedling growth Increased activity of α-amylase and content of soluble sugars Increased content of total phenolics, flavonoids, and GB Increased activity of SOD, POD, APX, and GPX Increased activity of enzymes involved in PA synthesis: ADC, ODC, and SAMDC, and enhanced expression of corresponding genes	[60]
*Oryza sativa* L. cv. Gobindobhog	Salinity	Spd Spm	Spm and Spd priming decreased Put and free Spm and Spd content in stressed seedlings Increase expression of genes of PA biosynthesis (*SAMDC*, *SPDS*, and *SPMS*) and PA catabolism (*PAO*) enzymes Increased expression of genes encoding antioxidant enzymes (CAT, GR, SOD, APX, and anthocyanin synthase in shoot, and CAT, GR, and SOD in root) Enhanced expression of genes encoding osmolytes (*P5CS* and *BADH1*) and enzymes of ABA biosynthesis pathway with the simultaneous increase in gene expression of ABA-inducible transcription factors and LEA in the seedling Reestablish expression of gene encoding RuBisCO small subunit (*RbcS*)	[50]
*Oryza sativa* L. cv. IR-64 (sensitive) and Nonabokra (tolerant)	Salinity	Spd Spm	Decreased content of Put, Spd, and Spm in Nonabokra and increased of Spm and Spd in IR-64 cultivars Increased expression of genes encoding CAT, SOD, APX, and GR with the differences between salt-sensitive and tolerant cultivars Impacted expression of genes encoding enzymes involved in osmolyte synthesis (*BADH1*, *P5CS*, and *PDH*), PA biosynthesis (*SAMDC*, *SPDS*, and *SPMS*), PA catabolism (*DAO* and *PAO*), and salt stress-related proteins (*NHX1*, *NCED3*, *TRAB-1*, *WRKY-71*, and *LEA*) Increased expression of *RbcS*	[51]
*Oryza sativa* L. cv. viz., Huanghuazhan (HHZ, inbred) and Yangliangyou6 (YLY6, hybrid)	Drought	Spd	Improved seed germination and seedling growth Decreased content of MDA Increased level of soluble sugars Increased activity of CAT, POD, SOD Increased activity of α-amylase	[61]
*Oryza sativa* L.	Chilling	Spd	Improved growth of seedling shoots and roots Lower EL Enhanced activity of CAT, SOD, and APX in the seedling leaves	[62]
*Oryza sativa* L. cv. Niewdam (tolerant) and KKU-LLR-039 (sensitive)	Salinity	Spd	Improved seed germination and seedling growth Increased content of photosynthetic pigments Decreased content of Pro Increased content of phenolic compounds Altered Na^+^/K^+^ ratios with decreased of the toxic Na^+^ Increased antioxidant activity	[46]

## 4. Conclusions and Perspectives

Polyamines play an important role in seed biology as they are involved in the processes of embryo development, seed ripening, and germination, and can be used as a priming agent to enhance seed performance. PA seed priming enhanced stress tolerance, which is manifested through improved seed germination and seedling growth. Numerous studies have shown that PA-primed seeds exhibit improved growth parameters, including increased photosynthetic pigment content, reduced oxidative damage, and enhanced antioxidant enzyme activity. These biochemical changes are accompanied by alterations in gene expression related to stress responses, antioxidant defense, osmolyte production, and PA metabolism. Some studies have highlighted the role of Spd and Spm priming in promoting photosynthesis, reducing lipid peroxidation, and regulating ROS levels (Figure 2). PAs directly interact with DNA, influencing chromatin condensation, histone acetylation and deacetylation, and transmethylation of DNA. These interactions suggest a potential role for PAs in regulating the epigenetic control of gene expression. Therefore, future research should explore the involvement of PAs in priming memory through epigenetic mechanisms. Molecular changes, such as the modulation in gene expression of PA biosynthesis and degradation enzymes, further contribute to the enhanced stress tolerance observed in primed plants. However, despite the growing body of research, the precise molecular mechanisms underlying PA priming, particularly its effect on gene expression, remain unclear. Further in-depth analyses comparing PA priming with hydro-priming are required to fully elucidate the role of PAs in the complex gene expression regulatory network involved in plant stress responses. While some published studies have confirmed the direct role of PAs in regulating translation in plants, further research is needed to validate this mechanism’s contribution to plant development and seed physiology under both normal and stress conditions. Given the global imperative to enhance food production sustainably, more efforts should focus on translating research findings into practical applications.

## Figures and Tables

**Figure 1 ijms-25-12588-f001:**
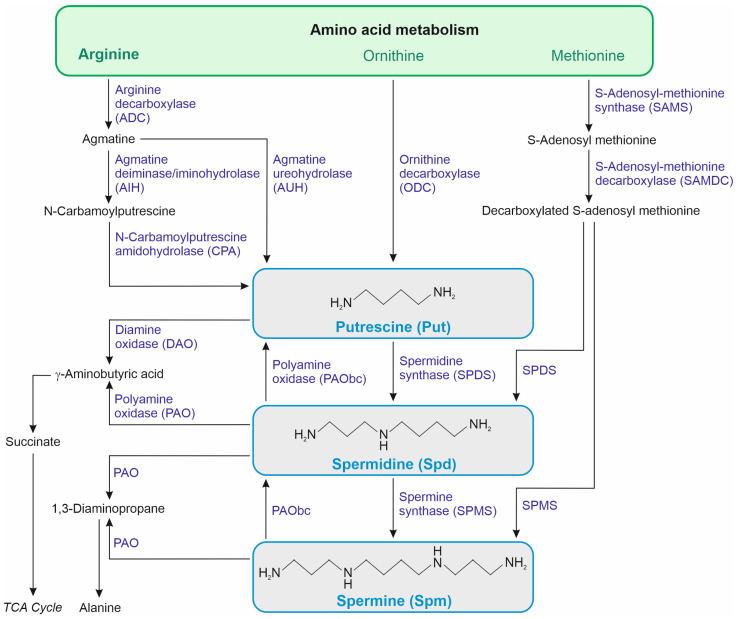
Simplified scheme of putrescine, spermidine, and spermine metabolism in plants.

**Figure 2 ijms-25-12588-f002:**
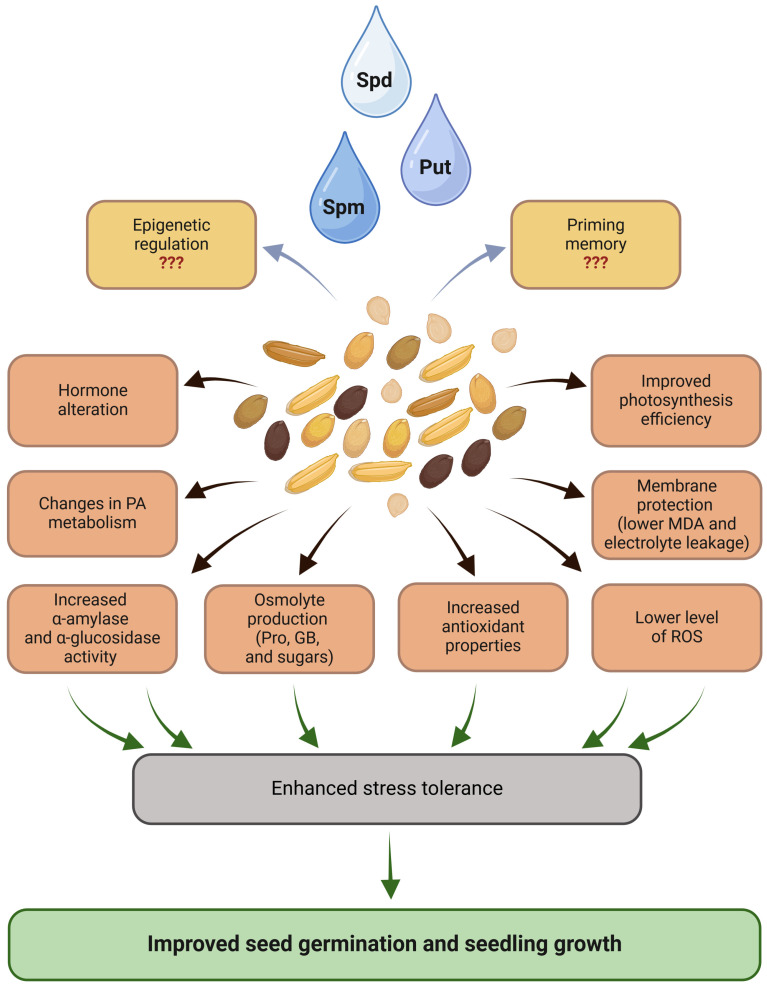
The general overview of the mechanisms underlying stress response upon PA seed priming. Abbreviations are explained in the text. The question marks refer to hypothetical PA-mediated epigenetic regulation and priming memory. Created in BioRender.

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
