# Peer review of "Polyamine Seed Priming: A Way to Enhance Stress Tolerance in Plants"

_ijms, 2024, doi:10.3390/ijms252312588_

Round 1

Reviewer 1 Report

Comments and Suggestions for Authors

Wojtyla et al. did a great job in reviewing the current state of knowledge on polyamine seed priming and further plant stress tolerance. The authors pointed out that these benefits are rooted in biochemical, physiological, and molecular mechanisms and showed much evidence from various studies. Some points could be more specific and further explained. For example, while molecular mechanisms of polyamine action are discussed, it would be advantageous to clearly indicate how such mechanisms would relate to specific physiological outcomes observed. More important, a clearer indication of future research directions is needed, as often repetition of established findings is made without detailed discussion of any new insights. While the different stress conditions summarized are superbly done, the discussion on various techniques of priming would benefit from further elaboration, possibly through a comparative analysis.

The review could be more impactful in its content by giving a more critical overview of the existing data on polyamine seed priming.
As it stands presently, much of the presentation is descriptive, with less attention given to discussing the gaps in literature or possible limitations of the methodologies of existing research. Improvement would be an in-depth discussion of the review on open questions on polyamine research, focusing on less-understood processes such as gene regulation, including priming memory. The figures are helpful, but detailed elaboration on metabolic pathways involved would really help explain and illustrate for the readers complex processes.

Specific changes:
Line 11: "Only in a few papers PA were described" → "PA have been described in only a limited number of studies".

Line 21: replace "has been an area of interest to humanity" by "has been of interest to researchers."
Line 36: "can be considered the first written mention of pre-sowing seed treatment" → "represents the earliest documented mention of pre-sowing seed treatment".
52 "Seed .. osmolytes" → "Seed priming induces osmolyte accumulation".
56 "depending .. used" → "depending on the plant species and the applied priming method".
69 replace "commonly" with "widely"

Author Response

Comments 1: Some points could be more specific and further explained. For example, while molecular mechanisms of polyamine action are discussed, it would be advantageous to clearly indicate how such mechanisms would relate to specific physiological outcomes observed.

Response 1: We have added more specific information about the molecular mechanisms of polyamine action and included additional details to clarify the connection between these molecular mechanisms and physiological outcomes (lines 116–138).

Comments 2: More important, a clearer indication of future research directions is needed.

Response 2: We have highlighted some of the most important areas for studying the impact of polyamine seed priming (lines 93-95, 323-325, conclusion section).

Comments 3: The discussion on various techniques of priming would benefit from further elaboration, possibly through a comparative analysis.

Response 3: The comparative analysis of various priming techniques is highly intriguing, as it could help identify the most effective methods that are easier to implement in practice. However, in the case of polyamines, seeds are treated through imbibition in a polyamine solution, with differences limited to the concentration of polyamines and the duration of priming or incubation. Variations in methods also stem from the wide range of species studied, making such comparisons unjustified. For this reason, we chose to focus not on differences between priming methods but rather on the effects exerted by polyamines themselves.

Comments 4: The review could be more impactful in its content by giving a more critical overview of the existing data on polyamine seed priming. As it stands presently, much of the presentation is descriptive, with less attention given to discussing the gaps in literature or possible limitations of the methodologies of existing research. Improvement would be an in-depth discussion of the review on open questions on polyamine research, focusing on less-understood processes such as gene regulation, including priming memory. The figures are helpful, but detailed elaboration on metabolic pathways involved would really help explain and illustrate for the readers complex processes.

Response 4: We acknowledge that we have not fully explored the mechanisms underlying the direct effects of polyamines on germination biology and stress resistance. However, we wish to emphasize that the primary goal of this work is to compile and organize information on the application of polyamines in seed priming. To the best of our knowledge, no review has yet been published that specifically addresses the significance and impact of seed priming with polyamines on seed germination, seedling growth, and stress resistance. Additionally, we have made slight modifications to Figure 2 to ensure greater coherence with the conclusion section.

Comments 5: Specific changes:
Line 11: "Only in a few papers PA were described" → "PA have been described in only a limited number of studies".
Line 21: replace "has been an area of interest to humanity" by "has been of interest to researchers."
Line 36: "can be considered the first written mention of pre-sowing seed treatment" → "represents the earliest documented mention of pre-sowing seed treatment".
52 "Seed .. osmolytes" → "Seed priming induces osmolyte accumulation".
56 "depending .. used" → "depending on the plant species and the applied priming method".
69 replace "commonly" with "widely"

Response 5: All specific changes have been applied to the text.

Reviewer 2 Report

Comments and Suggestions for Authors

The review is very interesting since the authors have reported the recent literature on the subject to give a comprehensive picture of the effects of seed priming with polyamines. Certainly the subject is of scientific and agronomical interest and the review should be published.

However, some changes should be made for a better understanding of the manuscript.

‘Introduction’ section should be added at line 20 (and 1. Seed priming………. deleted) .

 Then, in my opinion, the whole first part from Theophrastus to Darwin could be eliminated because, although having a historical interest, such information is not very pertinent for this scientific review on polyamines in priming.

Moreover, in section 2 and section 3 some arguments are repeated (for example section 2 lines 113-135 and section 3 lines 204-219 for antioxidant response, ROS and antioxidant enzymes.

Section 1 lines 136-149 and section 2 lines 195-203 for photosynthesis

Section 1 lines 150-157 and section 2 lines 259-268 for phytormones,

and in general also other aspects are repeated in the two sections).

 The authors could divide section 2 and 3 in subsections dedicating each subsection to a specific aspect of PA priming, for example on physiological effects on germination and seedling growth, on the production of ROS and oxidative stress enzymes, on photosynthetic system, on phytohormones, on gene expression etc (this would avoid any repetitions in section 2 and 3) making the review clearer and more understandable. The manuscript needs to be revised, my suggestion is major revisions

Author Response

Comments 1: ‘Introduction’ section should be added at line 20 (and 1. Seed priming………. deleted)

Response 1: It has been done.

Comments 2: Then, in my opinion, the whole first part from Theophrastus to Darwin could be eliminated because, although having a historical interest, such information is not very pertinent for this scientific review on polyamines in priming. 

Response 2: We accept the point of view of the Reviewer 2. It has been done.

Comments 3: 

Moreover, in section 2 and section 3 some arguments are repeated (for example section 2 lines 113-135 and section 3 lines 204-219 for antioxidant response, ROS and antioxidant enzymes.

Section 1 lines 136-149 and section 2 lines 195-203 for photosynthesis

Section 1 lines 150-157 and section 2 lines 259-268 for phytormones,

and in general also other aspects are repeated in the two sections).

 The authors could divide section 2 and 3 in subsections dedicating each subsection to a specific aspect of PA priming, for example on physiological effects on germination and seedling growth, on the production of ROS and oxidative stress enzymes, on photosynthetic system, on phytohormones, on gene expression etc (this would avoid any repetitions in section 2 and 3) making the review clearer and more understandable. 

Response 3: We would like to thank the Reviewer for raising this important issue. However, our intention was to first present the mechanisms of polyamine action on plants in section 2 and briefly discuss the effects observed in plants in response to exogenously applied polyamines. In section 3, we focused solely on polyamine seed priming and the observed changes in plants in response to this treatment. In our opinion, combining these two sections would not improve the understanding of the phenomenon. To clarify this idea for all readers, we have added the following information at the beginning of section 3: “As the previous section refers to exogenously applied and endogenous PA as regulatory molecules that impact plant stress responses, this section focuses on the intricate relationship between PA seed priming and the defence responses of plants under stress conditions.”

Reviewer 3 Report

Comments and Suggestions for Authors

In this review article, the authors summarized the effects of polyamine seed priming on germination, seedling development, and response of plants to abiotic stresses. The previous studies, including very recent publications, were intensively searched, and the text is well-organized for the readers to understand the outline of PA seed priming.

The reviewer has only 2 minor comments as follows.

Lines 44 to 46

The authors say “Seed priming is distinguished from pre-soaking due to subsequent seed drying back into initial water content, which allows storing and transportation.” 

The reviewer is afraid that drying process might not be necessary required for seed priming technology, especially for biopriming with microbes, The following publications are the examples.

Trichoderma mediated seed biopriming augments antioxidant and phenylpropanoid activities in tomato plant against Sclerotium rolfsii. Journal of Pharmacognosy and Phytochemistry 2019; 8(3): 2641-2647

Biopriming of maize germination by the plant growth-promoting rhizobacterium Azospirillum lipoferum CRT1. Journal of Plant Physiology 237 (2019) 111–119

Seed biopriming with P- and K-solubilizing Enterobacter hormaechei sp. improves the early vegetative growth and the P and K uptake of okra (Abelmoschus esculentus) seedling. PLoS ONE 15(7): e0232860. https://doi.org/10.1371/journal.

If hydro-priming (line 287) and water-priming (line 314) is the same, please use the same term.

Author Response

Comments 1: The reviewer is afraid that drying process might not be necessary required for seed priming technology, especially for biopriming with microbes.

Response 1: The authors thank the Reviewer for pointing this out. In the case of biopriming, the methodology does not fully align with the definition of priming. Moreover, since the term "biopriming" refers not only to microorganisms but also to secondary metabolites, the process of drying seeds back to their initial water content is present in some studies on seed biopriming. We have added a brief explanation regarding biopriming in the text: “However, among different methods described as biopriming, some of them involve seed imbibition together with bacterial inoculation of seed without subsequent drying” (lines 50-52).

Comments 2: If hydro-priming (line 287) and water-priming (line 314) is the same, please use the same term.

Response 2: It has been corrected.

Round 2

Reviewer 2 Report

Comments and Suggestions for Authors The authors have appropriately revised the manuscript by accepting some suggestions or specifying in the text the choice of the organization of sections 2 and 3. They have also improved some parts, for example the role of PA in the regulation of gene expression and other aspects enhancing the manuscript. The work can be published in this form